# A Novel Image-Based Diagnosis Method Using Improved DCGAN for Rotating Machinery

**DOI:** 10.3390/s22197534

**Published:** 2022-10-04

**Authors:** Yangde Gao, Farzin Piltan, Jong-Myon Kim

**Affiliations:** 1Department of Electrical, Electronics and Computer Engineering, University of Ulsan, Ulsan 44610, Korea; 2PD Technology Cooperation, Ulsan 44610, Korea

**Keywords:** rotating machinery, fault classification, deep convolutional generative adversarial networks

## Abstract

Rotating machinery plays an important role in industrial systems, and faults in the machinery may damage the system health. A novel image-based diagnosis method using improved deep convolutional generative adversarial networks (DCGAN) is proposed for the feature recognition and fault classification of rotating machinery. First, vibration signal data from the rotating machinery is transformed into time–frequency feature 2-D image data by a continuous wavelet transform and used for fault classification with the neural network method. The adaptive deep convolution neural network (ADCNN) is then combined with the generative adversarial networks (GANs) to improve the performance of the feature self-learning ability from input data. Compared with different fault diagnosis methods, the proposed method has better performance for image feature classification in rotating machinery.

## 1. Introduction

Rotating machinery creates power for autonomous systems and is widely applied in modern industry, such as energy systems, transportation, and engine systems. Rotating machinery works under complex environments for long periods of time, which may cause damage to machinery components [1]. Therefore, fault diagnosis technologies are required to analyze condition information and improve the operation reliability of rotating machinery.

Traditional methods, including wavelet decomposition and empirical mode decomposition, are widely applied in feature analysis for rotating machinery; however, there are some drawbacks that influence their performances [2].

Machine learning based methods are widely applied in the feature recognition of rotating machinery; for example, support vector machines (SVMs) use statistical learning theory and can achieve reasonable accuracy prediction for feature information; however, SVM has reduced performance in the case of processing large amounts of data with high noise information, and machine learning methods do not have such self-learning ability after presetting parameters [3,4,5].

Deep learning methods can achieve self-learning from training data and improve the prediction performance for nonlinear operation. Therefore, deep learning methods are one of the most popular technologies in data processing. For example, the convolutional neural network (CNN) method can change weights to adjust for its learning ability and achieves flexible structure for fault diagnosis [6,7,8,9]. To improve the self-learning ability of CNN, adaptive deep convolution neural network (ADCNN) methods have been applied to the training and testing of large data, identifying the target states for monitoring systems [10,11]. 

To improve the recognition performance with limited data, the generative adversarial network (GAN) method is applied to expand the original feature information and generative multiple synthetic information for neural networks. The process can develop different branches using a real sample to generate similar fake samples, which can improve the self-learning ability and classification recognition [12,13,14,15]. However, to maintain a balance between the two branch networks of the GAN architecture, the process may be difficult because the gradient disappears. As a result, the enhanced GAN was proposed to improve the synthesis quality [16,17]. A framework using deep convolutional generative adversarial networks (DCGAN) is proposed for solving model collapse and generating a high-quality process for data [18,19,20]. 

To further improve self-learning for training and testing, ADCNN is combined with DCGAN to develop a novel, multiple deep convolutional generative adversarial network (MDCGAN) for fault diagnosis in rotating machinery. In summary, the main contributions are as follows:(1)A novel image diagnosis method using improved DCGAN is proposed for the feature recognition and fault classification in rotating machinery to ensure the safety of the operation.(2)In the proposed method, vibration signal data from the rotating machinery is transformed into time–frequency feature 2-D images data by a continuous wavelet transform (CWT) with time–frequency characteristics for vibration signals. The vibration signal data are better for training and testing neural networks than 1-D dimensional signals.(3)ADCNN is combined with GANs that have two neural networks, a generative G and a discriminator D, to improve the self-learning ability of the architecture. An improved DCGAN architecture was developed by adding different layers to different parts of a generative G and a discriminator D.(4)Experiments using data from the Ulsan Industrial Artificial Intelligent (UIAI) Laboratory were used to verify the proposed method, which had better performance for image feature classification in rotating machinery than other fault diagnosis methods.

The rest of the paper is organized as follows. Section 2 and Section 3 provide a theoretical background. Detailed information on the proposed method is described in Section 4. The experimental result and discussion are shown in Section 5 and Section 6, respectively. Finally, Section 7 includes the conclusions.

## 2. Continuous Wavelet Transform

The original data from vibration signals is a 1-D signal, which only shows amplitude information and is not suitable for process by deep learning methods, so before training and testing, The vibration signal is preprocessed and representative features for deep learning methods are extracted to bear health state classification. A defect in the bearing produces shocks, resulting in complex variations in terms of amplitude and the distribution of impulses in the vibration signals. Utilizing the complex variation in the time series vibration signal allows a better understanding of the health state of the system under study, such as bearings. Entropy is one of the powerful metrics to evaluate these kinds of complexities in the signal. Landauskas et al. [8] constructed the pattern of permutation entropy as a function of time delay, which offers a way to unveil the presence of structures in the vibration signal due to bearing health conditions on multiple temporal scales in the form of 2D images. The method provides a precise description of complex nonstationary signals, however, from the literature, it is evident that the permutation entropy calculated as a function of the time delay does not take information regarding the amplitudes [21]. 

To utilize the variations in the vibration signals due to the bearing working conditions in this study, continuous wavelet transform with source wavelet Morse having the symmetry parameter = 3, is used. For complete details about the Morse wavelet, readers are advised to refer to [22]. The continuous wavelet transform (CWT) is used to transform vibration signals and obtain 2-D time-frequency images; 2-D images not only show the frequency energy information immediately, but also have some advantages for deep methods. The wavelet transform makes use of basic functions, and the variable window to remove noises effectively from raw signals with noises, so CWT is more suitable for nonstationary vibration signals. Meanwhile, the transform can extract important accurate information from the vibration signals and achieves feature localization in time-frequency energy, so the CWT method is more effectively than other signal analysis methods [4]. 

The continuous wavelet transform (CWT) method uses a wavelet function to analyze the vibration signals x(t) and ensures the correlation information that is used for analysis. The process is defined by the following formulate [4,7]:(1)$$Cw\left(a,b\right)=\int_{-\infty }^{+\infty }x\left(t\right)\psi_{a,b}*dt\;=\frac{1}{a}\;\int_{-\infty }^{+\infty }x\left(t\right)\psi \;*\frac{\left(t-b\right)}{a}\;dt$$
where ψ is the point continuous mother wavelet, a and b are the parameters of the wavelet function, the a denotes the scale and b makes sure the translation for basis wavelet, these parameters have continuous characteristics and used for wavelet transform. The CWT method also can show the energy information, $\psi^{*}\left(t\right)$ denotes the complex conjugate from the wavelet basis and the $1/\sqrt{a}$ represents the energy-normalized coefficient, the finite amount energy of the wavelet function can be described as follow:(2)$$E=\int_{-\infty }^{+\infty }{\left|\psi \right|}^{2}dt<\infty \;\;C_{\psi }=\int_{-\infty }^{+\infty }\frac{{\left|\varphi \left(w\right)\right|}^{2}}{w}dw<\infty$$
where the φ(w) is the Fourier transform and described as follow:(3)$$\varphi \left(w\right)=\int_{-\infty }^{+\infty }\psi \left(t\right)e^{-iwt}dt$$
where w=2πf is the circular frequency, the CWT can show the energy information on the spectrum from vibration signals.

## 3. Introduction of Adaptive Deep Learning Neural Networks

ADCNN evolves from the basic theory of CNNs, mainly using convolution layers, pooling layers, several fully connected layers, and one final output layer, to achieve feature extraction from input data [11].

### 3.1. Convolution Layer

The convolution layers use kernels to adjust interactions and the convolutional operation, the convolutional process can filter redundancy information from the input data and extract feature information for the next layer [8]. The whole convolution process is shown in the following function:(4)$$\;x_{n}^{m}=f\left(\underset{i\in K_{n}}{\sum }x_{i}^{m-1}*w_{in}^{m}+b_{n}^{m}\right)$$
where $x_{n}^{m}$ denotes the result of the *m*th kernel in the convolutional layer n, $K_{n}$ is the convolution region, $w_{in}^{m}$ is the weight in convolutional layer in, $b_{n}^{m}$ denotes the bias vector, and f(∗) is a nonlinear activation function.

### 3.2. Pooling Layer

After the convolutional layer with convolutional operation, the max pool layer uses the maximum value to reduce the dimensions and extract useful feature information from a previous output [5]. This architecture can also reduce the risk of overfitting. The whole process is described as follows:(5)$$x_{n}^{m}=f(w_{n}^{m}*\max(x_{n}^{m-1})+b_{n}^{m})$$
where max(∗) represents the max pooling function, $w_{n}^{m}$ denotes the weight, $x_{n}^{m}$ is the output, and $b_{n}^{m}$ is the bias. 

### 3.3. Full Connected Layer

At the end of CNN, a full connected layer is usually used to map feature information into the final classifier, the specific calculation process can be described as:(6)$$y^{z}=f\left(w^{z}x^{z-1}+b^{z}\right)$$
where $y^{z}$ is the output for the final connected layer and z is the network. 

### 3.4. Backward Propagation

The loss function process is determined by the cross-entropy loss formula, and it is an important aspect of a neural network that makes it feasible for use as a gradient of the error function to achieve updated weights and biases for the prediction formular, and finally optimize the global target parameters from input data information [10]. The cross-entropy loss formula is expressed as:(7)$$E\left(w\right)=\frac{1}{n}\overset{n}{\underset{z=1}{\sum }}[y_{z}ln\bar{y_{z}}+\left(1-y_{z}\right)\ln\left(1-\bar{y_{z}}\right)]$$
where $\bar{y_{z}}$ represents the predicted value and $y_{z}$ is the actual target. 

## 4. The Modified DCGAN

The DCGAN architecture is developed from the GAN and CNN. A traditional GAN mainly has two parts, a generator (G) and a discriminator (D), as shown in Figure 1. Generative G data from random noise, is then combined with the real data to be used in D; these models use minimax adversarial training to improve both networks’ circulation [15]. The object formular is described as follows:(8)$$\underset{D}{\max L_{D}}=E_{x\sim P_{da}}\left[logD\left(x\right)\right]+E_{z\sim p_{z}}\left[\log\left(1-D\left(G\left(z\right)\right)\right)\right]\;\underset{G}{{\;\mathrm{minL}}_{G}}=E_{z\sim p_{z}}\left[\log\left(1-D\left(G\left(z\right)\right)\right)\right]$$
where $P_{da}$ is the x distribution, $p_{z}$ is the z distribution, D(x) is the possibility for real distribution, G(z) is the generated sample, and E is the expectation. 

The DCGAN model combines GAN with a CNN method. GAN includes two neural networks: a generative G and a discriminator D. Compared with a traditional GAN method, the networks are conditioned on different layers in the DCGAN architecture [18]. To improve DCGAN, a modified version is proposed where the two neural networks are changed by different layers with different parts of generative G and discriminator D. The improved DCGAN is then used to classify the 2-D image datasets for vibration signals and achieve high accuracy classification. The architecture is described in Figure 2.

In this improved DCGAN method, some parameters are optimized to reduce the collapse of the network architecture, which can take advantage of the distribution convolutional layers to generate feature information for training and testing. Before training, vibration signals are transformed into 2-D time–frequency feature images by CWT, which provides an overcomplete representation of a signal by the translation and scale, the main function is described as $Cw\left(a,b\right)=\int_{-\infty }^{+\infty }x\left(t\right)\psi_{a,b}*dt$, where ψ is the continuous mother wavelet, a is the scale, and b is for translation. The same process was previously described. To make the proposed method stable for training image data, batch normalization (BN) and ReLU activation function were used to optimize the weight parameters for the gradient process.

In the improved DCGAN, the convolution kernel was optimized to select the size for feature information, the size of the convolution kernel and stride was 3 × 3, and the filter was 16. The dropout was 0.02 in the generative model. Some layers of the generative mode are shown in Figure 3 and the structure information is illustrated in Table 1. 

In the discriminative model, the size of the convolution kernel and stride was 3 × 3, the filter was 16, and the dropout was 0.02, as shown in Figure 4. The structure information is described in Table 2. 

## 5. Experimental Results 

To verify the performance of the proposed method, the bearing tested platform was designed by the Ulsan Industrial Artificial Intelligent (UIAI) Laboratory of Ulsan University, South Korea [7]. As shown in Figure 5, the vibration bearing data includes four different conditions: inner race damaged bearings, outer race damaged bearings, roller race damaged bearings, and a normal bearing; the collected data can achieve classification for four different conditions by the proposed method. In this experiment, a speed of 1800 rpm was used for the three-phase motor and the sampling rate was 25 kHz for the vibration data.

FAG NJ206-3-TVP2 bearing is used during the experiment having a severe crack with a length 3 mm, a width of 0.3 mm, and a depth of 1mm on the outer race, roller, and inner race, respectively. The crack was created using an electronic machine called an electro-discharge machining; AE sensors (R151-AST type), and an accelerometer (PCB-622B01) were used to record vibration signals, and then a NI-9234 DAQ device was used to collect a large amount of data from the sensors. Table 3 provides details about the devices used in this experiment.

Figure 6 illustrates the frequency spectrum of the vibration signal obtained from the bearing under different operating conditions. The defect frequencies for inner race, outer race, and ball fault, are calculated using the formulation given in [23]. It can be seen from Figure 6a that when the bearing health condition changes from normal to defective condition, the fault frequency starts appearing in the vibration spectrum, as can be seen in Figure 6a–d. Thus, the collected vibration signals can be used for the fault diagnosis of the bearing. Therefore, the 1-D is further preprocessed and is transformed into 2-D time-frequency images by the CWT method; 2-D time-frequency images can show the time-frequency feature and energy feature, which are easier to identify. The improved DCGAN method was used for training and testing, then the classification results were calculated. The whole experiment process for the proposed method is described in Figure 7.

The transform results are described in Figure 8. The 1-D vibration signals can show the amplitude information, and these vibration signals also have some noises that influence feature recognition, with better analysis for the proposed method. The 1-D vibration signals were transformed into 2-D time-frequency images by the CWT method, and the 2-D time–frequency images are described in Figure 8B. These images can show the time frequency information that removes noise and shows the important energy information that is easier for localization, recognition, and classification. The input image size is 64 × 64. Then, the improved DCGAN was used to train and test the 2-D image data.

There are many redundancies in the 2-D image space; only energy features are used for classification. To describe the self-learning process of the improved DCGAN method, the convolutional process is used to extract features and reduce redundancy information. The convolutional process can filter noise signals and reduce high-dimensional features by each layer, and extraction features are easier with regards to recognition and classification.

After obtaining all 2-D image data by the CWT method, the 2-D images were processed by the improved DCGAN method. The proposed method makes use of the advantages of convolutional layers in the ADCNN method to extract the feature information and remove noise from the input 2-D images; the ADCNN method has a stronger self-learning ability that can filter redundancy information, so the feature information is easier to classify, and some processes showed results for the following conditions: normal, inner race fault, outer race fault, and roller race fault. The results for the convolutional process are described in Figure 9:

To describe the cluster ability process of the improved DCGAN method, the convolutional process can filter noise signals and reduce high-dimensional features by each layer, and the t-distributed stochastic neighbor embedding (t-SNE) algorithm is used to extract features and reduce redundancy information; it also shows a visualization with a two-dimensional plane.

To evaluate the performance of the proposed method a proper configuration of testing and training sets are made. As such, four different configurations of training and testing sets are prepared. The first configuration of the dataset contains 744 training samples and 496 testing samples, this configuration will be referred to as “A” in the text. A second configuration of the dataset is made, which contains 868 samples for training and 372 samples for testing, this configuration will be referred to as “B” in the text. For the third configuration, 992 samples were kept in the training set, while the remaining 248 samples were used for validation of the proposed model, this configuration will be referred to as “C” in the remaining parts of this study. For the fourth configuration, 1116 samples were used for training the proposed model and the remaining 124 samples were used for validating the proposed fault diagnosis method. Furthermore, in this study, t-distributed stochastic neighbor embedding (t-SNE) was used to show self-learning ability from the input 2-D image data. 

After applying the proposed method to dataset configuration A, the method classified the bearing operating conditions with a total prediction accuracy of 99%. The discriminant feature space obtained from the proposed method for dataset configuration A is presented in Figure 10. From Figure 10a it can be observed that, initially, the t-SNE with a two-dimensional feature space does not show separability between the features. However, with different layers step by step, the samples representing different bearing operating conditions were gradually separated with high between class distance and less interclass scatteredness, as can be seen from Figure 10b–d. This is the reason for the higher classification accuracy of the improved DCGAN. Figure 11 shows the confusion matrix for the proposed method. From the confusion matrix per class, prediction accuracies were calculated and are presented in Table 4. From Table 4 it can be observed that: the method classified the inner race fault with the accuracy of 97%; outer race fault with 100% accuracy; roll race fault with 100% accuracy; and normal bearing operating condition with 99% accuracy. The reference method ADCNN classified the bearing health conditions with a total prediction accuracy of 98.5%, which is slightly lower than the proposed method. For the inner race fault, the ADCNN achieved 96% accuracy; 98% for the outer race fault; 100% for the roll race fault; and 99% for normal operating conditions. Another reference method used for comparison is DCNN. The method DCNN classified the bearing health conditions with a total prediction accuracy of 97.9%, which is lower than the proposed method. For the inner race fault, the DCNN achieved 94% accuracy; 99% for the outer race fault; 100% for the roll race fault; and 99% for normal operating conditions. 

Continuously, the proposed method is applied to dataset configuration B and the performance was evaluated. From Table 4 it can be seen that the method classified the bearing operating conditions with a total prediction accuracy of 98.9%. The discriminant feature space obtained from the proposed method for dataset configuration B is presented in Figure 12. From Figure 12a it can be observed that, initially, the features have no discriminancy. However, as the sample goes through the layers of the improved DCGAN, the discriminancy of the features representing different bearing operating conditions improves, as can be seen from Figure 12b–d. Figure 13 shows the confusion matrix obtained for dataset configuration B. From Table 4 it can be observed that the method classified the inner race fault, outer race fault, roll race fault, and normal bearing operating condition with the accuracy of 97%, 99%, 100, and 99%, respectively. The reference method ADCNN classified the bearing health conditions with a total prediction accuracy of 98.3%, which is lower than the proposed method. For the inner race fault, outer race fault, roll race fault, and normal bearing operating condition, the ADCNN achieved 94%, 100%, 98%, 100, and 94 accuracies. For the reference method DCNN, a total prediction accuracy of 97.8% was obtained, which is lower than the proposed method. For the inner race fault, the DCNN achieved 91% accuracy, 99% for the outer race fault, 100% for the roll race fault, and 100% for normal operating conditions. 

Similarly, the proposed method is applied to dataset configuration C. The method classified the bearing operating conditions with a total prediction accuracy of 98.9%. The discriminant feature space obtained from the proposed method for dataset configuration C is presented in Figure 14. From Figure 14a it can be observed that, initially, the t-SNE with a two-dimensional feature space does not show separability between the features. However, with different layers step by step, the samples representing different bearing operating conditions were gradually separated with high between class distance and less interclass scatteredness, as can be seen from Figure 14b–d. This is the reason for the higher classification accuracy of the improved DCGAN. Figure 15 shows the confusion matrix for the proposed method. From Table 4 it can be observed that the method classified the inner race fault with the accuracy of 100%, outer race fault with 95% accuracy, roll race fault with 100% accuracy, and normal bearing operating condition with 100% accuracy. The reference method ADCNN classified the bearing health conditions with a total prediction accuracy of 98%, which is lower than the proposed method. For the inner race fault, the ADCNN achieved 100% accuracy, 99% for the outer race fault, 99% for the roll race fault, and 100% for normal operating conditions. Another reference method used for comparison is DCNN. The method DCNN classified the bearing health conditions with a total prediction accuracy of 97.9%, which is lower than the proposed method. For the inner race fault, the DCNN achieved 94% accuracy, 100% for the outer race fault, 100% for the roll race fault, and 97% for normal operating conditions. 

Continuously, the proposed method is applied to dataset configuration D and the performance was evaluated. From Table 4 it can be seen that the method classified the bearing operating conditions with a total prediction accuracy of 99.2%. The discriminant feature space obtained from the proposed method for dataset configuration D is presented in Figure 16. From Figure 16a it can be observed that, initially, the features have no discriminancy. However, as the sample goes through the layers of the improved DCGAN the discriminancy of the features representing different bearing operating conditions improves, as can be seen from Figure 16b–d. Figure 17 shows the confusion matrix obtained for dataset configuration D. From Table 4 it can be observed that the method classified the inner race fault, outer race fault, roll race fault, and normal bearing operating condition with the accuracy of 100%, 100%, 100%, and 97%, respectively. The reference method ADCNN classified the bearing health conditions with total prediction accuracy of 98.4%, which is lower than the proposed method. For the inner race fault, outer race fault, roll race fault, and normal bearing operating condition, the ADCNN achieved 94%, 100%, 100%, 100%, and 97% accuracies. For the reference method DCNN a total prediction accuracy of 97.8% was obtained, which is lower than the proposed method. For the inner race fault, the DCNN achieved 92% accuracy, 100% for the outer race fault, 100% for the roll race fault, and 98% for normal operating conditions. For a clearer understanding, the total prediction accuracies of the proposed and reference methods are presented in Figure 18.

To validate the sensitivity of the proposed method towards the incipient defects in the bearing, the Case Western University (CWU) bearing dataset is utilized. The CWU dataset and its configuration is used to validate the sensitivity of the proposed method towards incipient faults and can be explained as follows: the vibration signals were collected from the drive-end bearing with a sampling frequency of 12kHz under normal conditions. For outer race, ball, and inner race defect conditions, a crack of size 0.17 mm is created in the bearing and vibration signals with a sampling frequency of 12kHz were collected. During the experiment, the shaft speed was kept at 1797rpm. For a detailed description of the dataset, readers are advised to study [24]. Before applying the proposed method to the CWU dataset, a proper configuration of the dataset was made. As such, the data were divided into 80% of training data and the remaining 20% of the samples were used for validation purposes. After applying our proposed method to the CWU dataset we obtained 100% classification accuracy for normal conditions. The method classified the inner race fault, outer race fault, and ball defect with the accuracy of 97%, 99%, and 97%, respectively. The overall classification accuracy achieved by the proposed method for incipient bearing defects is 98.25%. The overall classification accuracy of the proposed method for incipient and severe faults is above 95%, which illustrates that the proposed method is sensitive to varying severity defects in the bearings, irrespective of the shaft speed. 

The proposed method classified the bearing defects effectively under macrostructural vibration noise. The signal-to-noise ratio (SNR) for the inner race fault, ball fault, and outer race fault is calculated. The SNR for the inner race fault, ball fault, and outer race fault was −6.80, −5.74, and −3.84, respectively. From the SNR it can be concluded that the obtained vibration signals contain macrostructural vibration noise as the signal power is lower than the noise power. With such a low signal strength the proposed method classified the bearing health conditions with a prediction accuracy higher than the reference methods. The high classification accuracy of the proposed method shows that the proposed method is robust towards the bearing health condition classification irrespective of the vibration noise.

The GAN has two neural networks, a generative G and a discriminator D. Compared with traditional GAN, in the improved DCGAN architecture, two neural networks are changed by different layers through the different parts, which can improve the self-learning ability from input data. The improved DCGAN takes advantage of the convolutional layers and GAN to improve better performance than the DCNN method and ADCNN method for fault classification in rotating machinery.

## 6. Discussion

The evaluation parameters show that the improved DCGAN has a good performance for fault classification in rotating machinery. 1-D vibration signals were collected from the vibration bearing platform; these vibration signals only show amplitude information and some noises influence analysis for deep learning methods. To remove the noise and to show more information to the vibration signals, the CWT method was used to transform 1-D signals into 2-D images. These 2-D images can show the time-frequency feature and also localize some important energy information that are useful for recognition, however some pace information in these 2-D images is redundancy for training and testing, so improved DCGAN makes full use of the convolutional process, which can remove noise and reduce high dimensional information form input 2-D data. The convolutional process also shows the performance from input 2-D images. Finally, the extracted features were used for training and testing. To show the self-learning ability of classification, the t-NSE method is used to show the cluster performance for different layers. The input images are not separated into different conditions clearly, however with the different layer step, the different data conditions are classified into four conditions. Meanwhile, the improved DCGAN makes use of the convolutional layers and the GAN method to classify the fault 2-D images. GANs includes two neural networks, a generative G and discriminator D. In the improved DCGAN architecture, the two neural networks were changed by different layers via different generative G and a discriminator D. The improved DCGAN improves the performance of the self-learning ability from the input data, and the confusion matrix shows the prediction for the improved method. Compared with the DCNN and ADCNN methods, the improved DCGAN has high prediction accuracy. However, some methods for signal processing need more work in the future. As we know, the CWT method is a classical method that can transform 1-D signals into 2-D image, however, for CWT the selection of the source wavelet is very important, as the mother wavelet is the basis for decomposition. To choose the proper source wavelet, a lot of experiments, as well as expertise in signal processing, are needed. Therefore, for signal preprocessing, in the future, instead of CWT a self-adaptive signal processing method can be used, such as EMD, and EEMD.

## 7. Conclusions

In this paper, a novel fault classification method, based on improved DCGAN, is proposed for rotating machinery. For the first time, 1-D vibration signals were transformed into 2-D time-frequency feature images via the CWT method. The 2-D images showed the energy for different time–frequency, and the dataset information was more suitable for training and testing in the proposed method.

To further show the self-learning ability of different layers for these datasets, the t-SNE method was used to show the performance from raw datasets for final classification by different layers, demonstrating that the image data were well classified with the proposed method.

GANs includes two neural networks, a generative G and discriminator D. Compared with a traditional GAN, in the improved DCGAN architecture, the two neural networks were changed by different layers via different generative G and a discriminator D, which means ADCNN with GAN improved the performance of the self-learning ability from the input data.

Finally, the improved DCGAN takes advantage of the convolutional layers and GAN to achieve better performance than the DCNN and ADCNN methods.

## Figures and Tables

**Figure 1 sensors-22-07534-f001:**
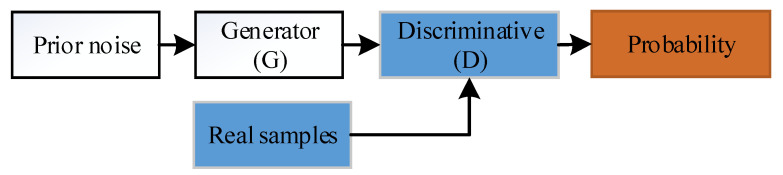
The GAN structure.

**Figure 2 sensors-22-07534-f002:**
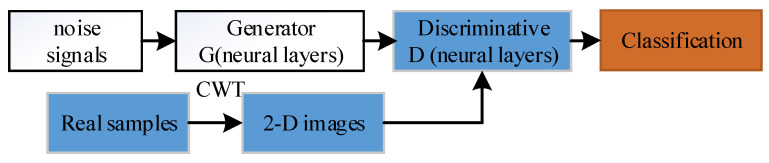
The architecture of the proposed method.

**Figure 3 sensors-22-07534-f003:**
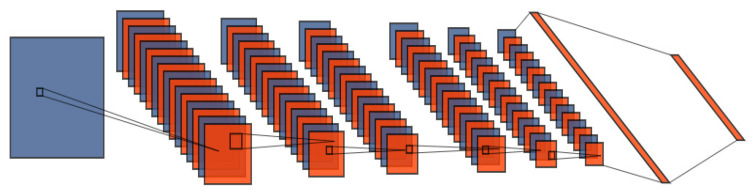
Construction of the generative model.

**Figure 4 sensors-22-07534-f004:**
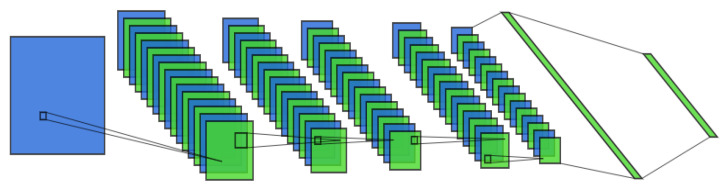
The architecture of the discriminative model.

**Figure 5 sensors-22-07534-f005:**
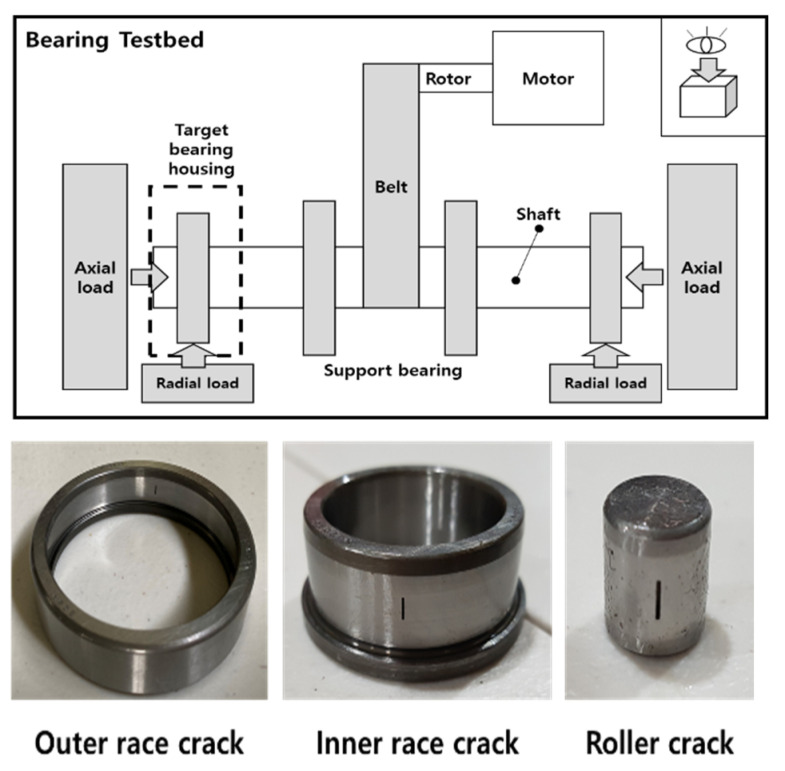
Images of the bearing test platform.

**Figure 6 sensors-22-07534-f006:**
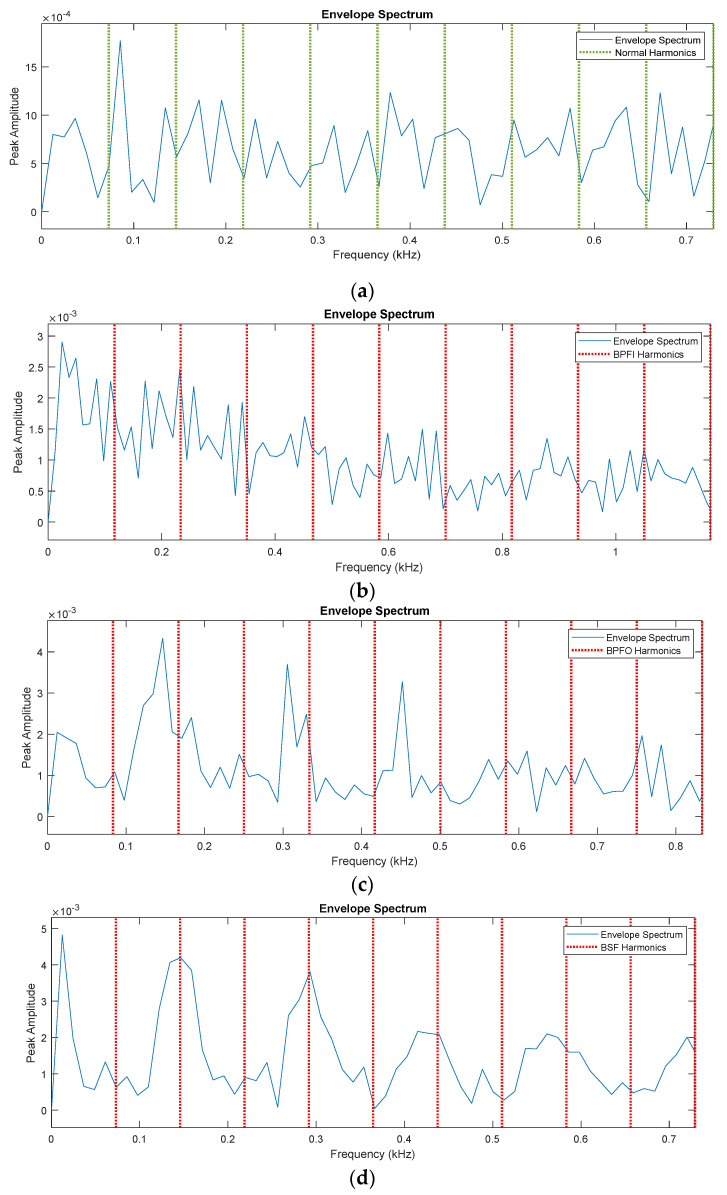
Envelope–spectrum for different bearing operating condition: (**a**) normal, (**b**) inner race fault, (**c**) outer race fault, (**d**) ball race fault.

**Figure 7 sensors-22-07534-f007:**
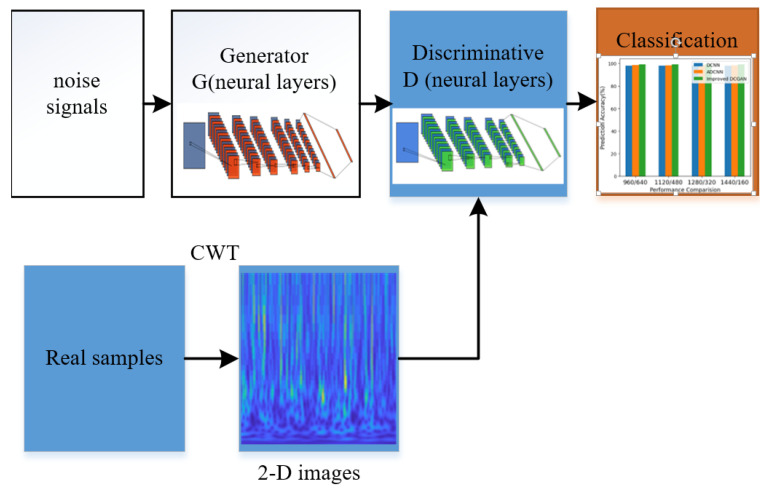
The whole experiment process for the proposed method.

**Figure 8 sensors-22-07534-f008:**
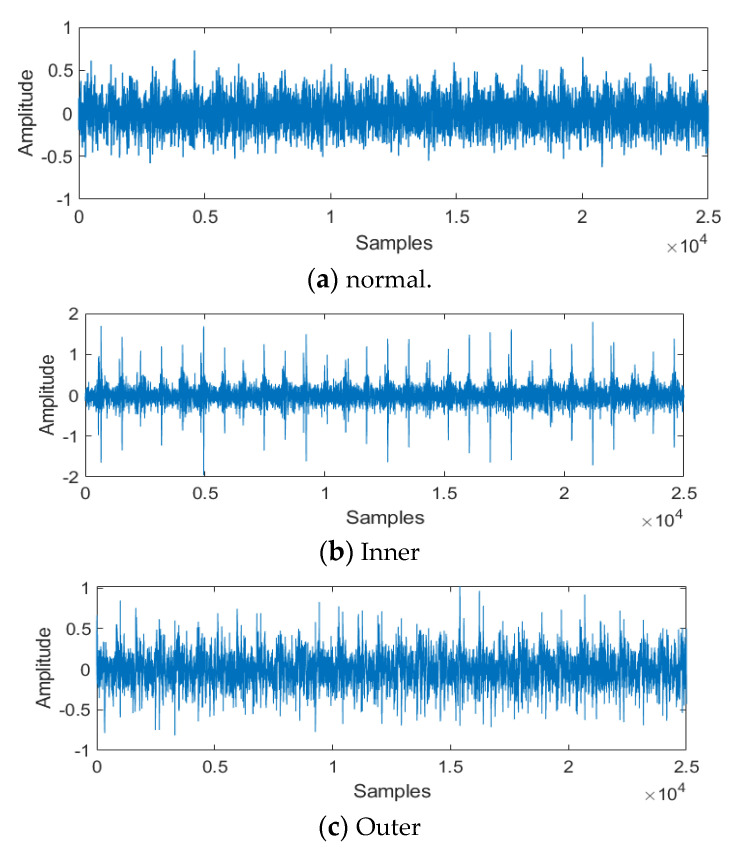
1−D vibration signals and 2−D time–frequency images for vibration signals: (**a**) normal, (**b**) inner race fault, (**c**) outer race fault, and (**d**) roller race fault.

**Figure 9 sensors-22-07534-f009:**
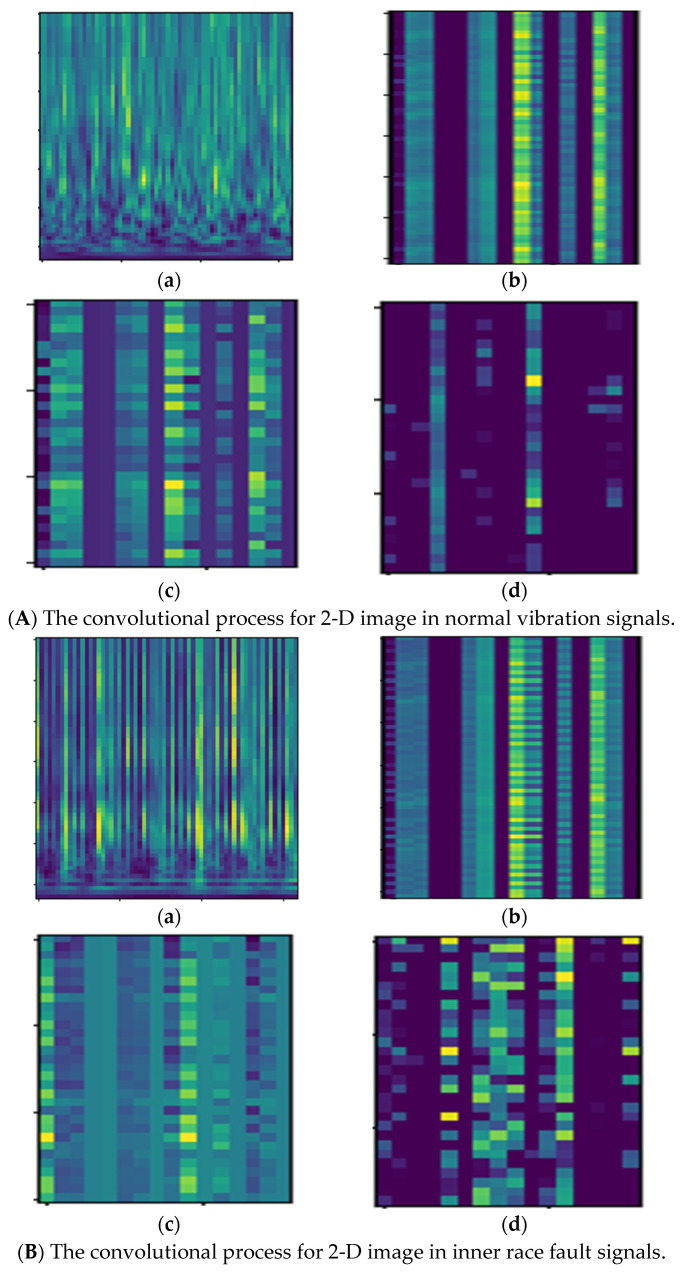
The visualization for convolutional process in 2–D images under different condition. (**a**) 2–D images for normal, inner race fault, roller race fault, outer race fault vibration, (**b**–**d**) visualization for convolutional process in (**A**–**D**).

**Figure 10 sensors-22-07534-f010:**
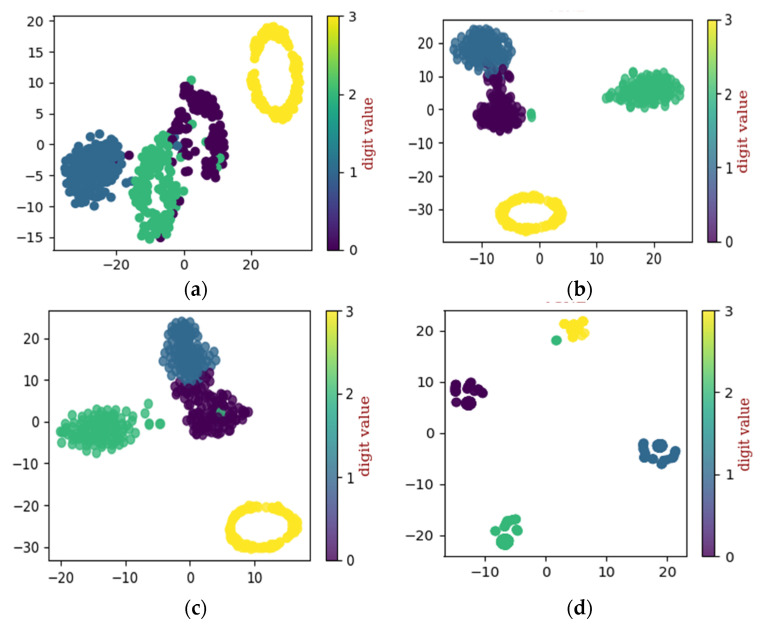
The t-SNE for self-learning ability utilizing dataset configuration A: (**a**) raw data, (**b**) convolution layer, (**c**) convolution layer, and (**d**) final classification.

**Figure 11 sensors-22-07534-f011:**
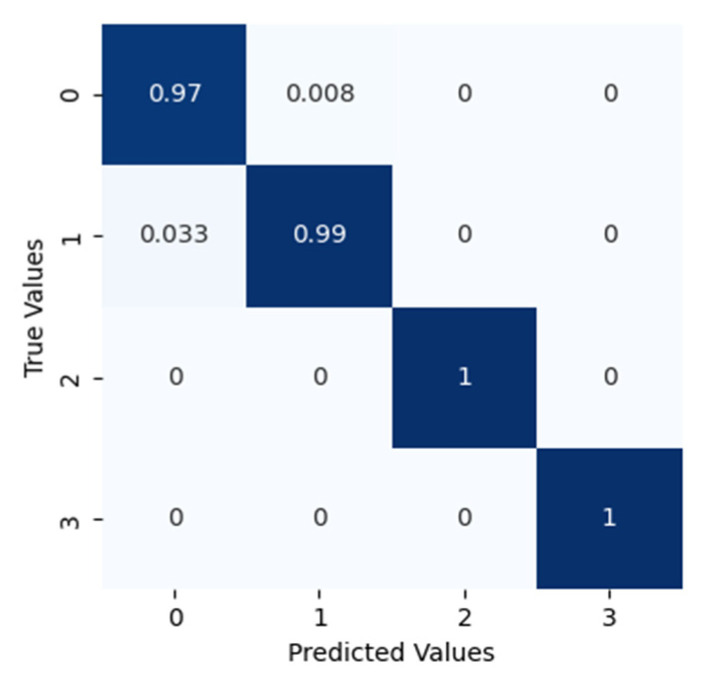
The confusion matrix for this classification.

**Figure 12 sensors-22-07534-f012:**
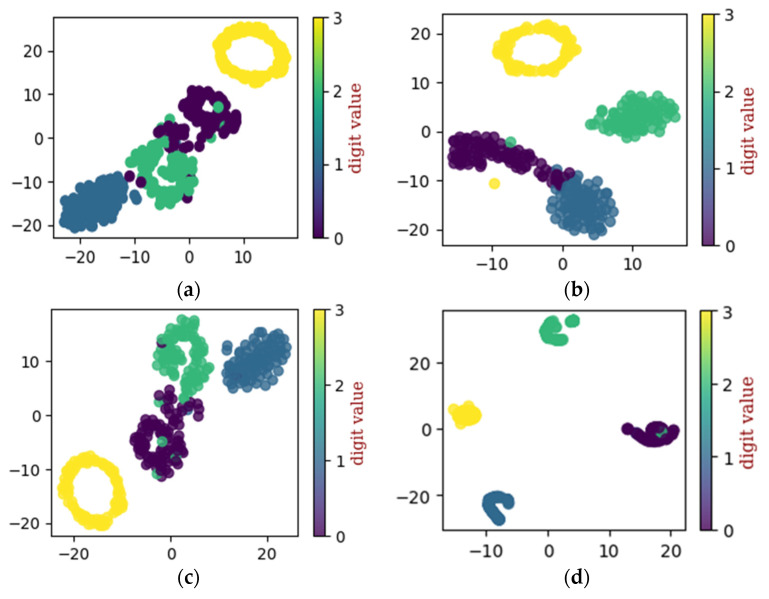
The t-SNE for self-learning ability utilizing dataset configuration B: (**a**) raw data, (**b**) convolution layer, (**c**) convolution layer, and (**d**) final classification.

**Figure 13 sensors-22-07534-f013:**
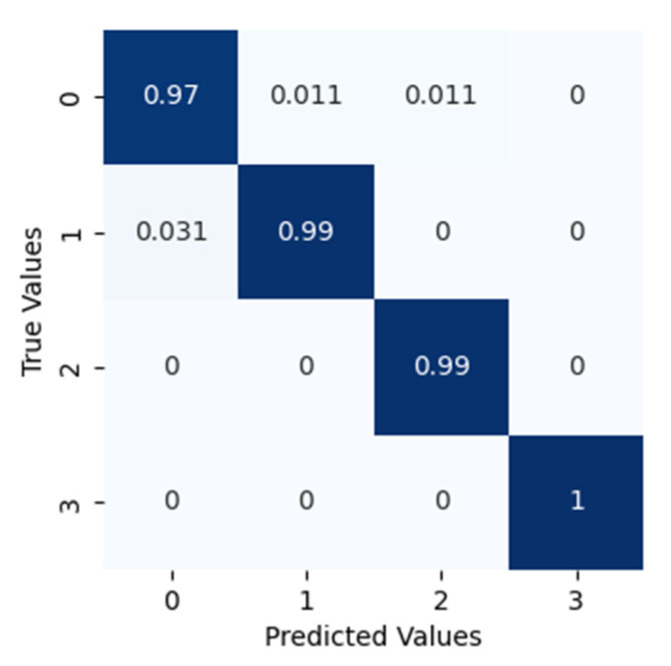
The confusion matrix for this classification.

**Figure 14 sensors-22-07534-f014:**
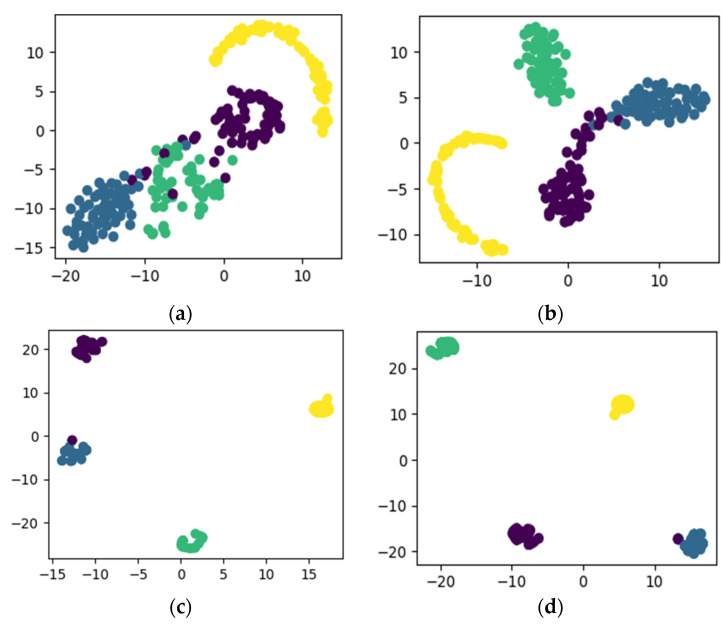
The t-SNE for self-learning ability utilizing dataset configuration C: (**a**) raw data, (**b**) convolution layer, (**c**) convolution layer, and (**d**) final classification.

**Figure 15 sensors-22-07534-f015:**
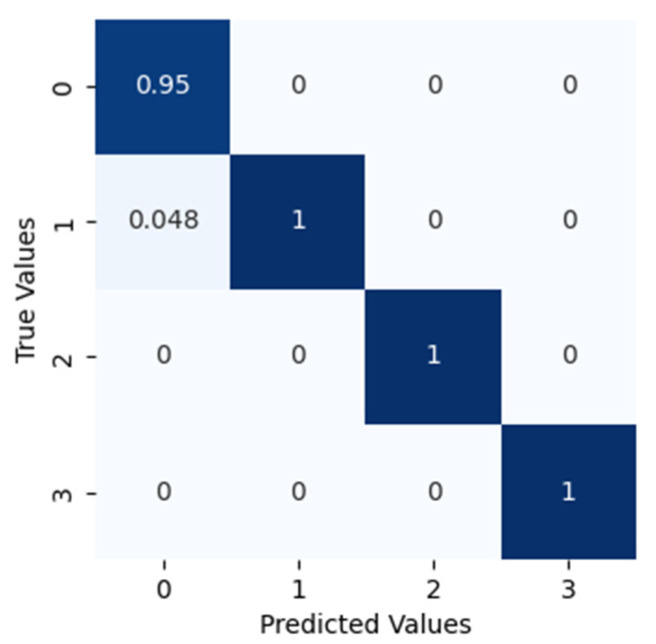
The confusion matrix for this classification.

**Figure 16 sensors-22-07534-f016:**
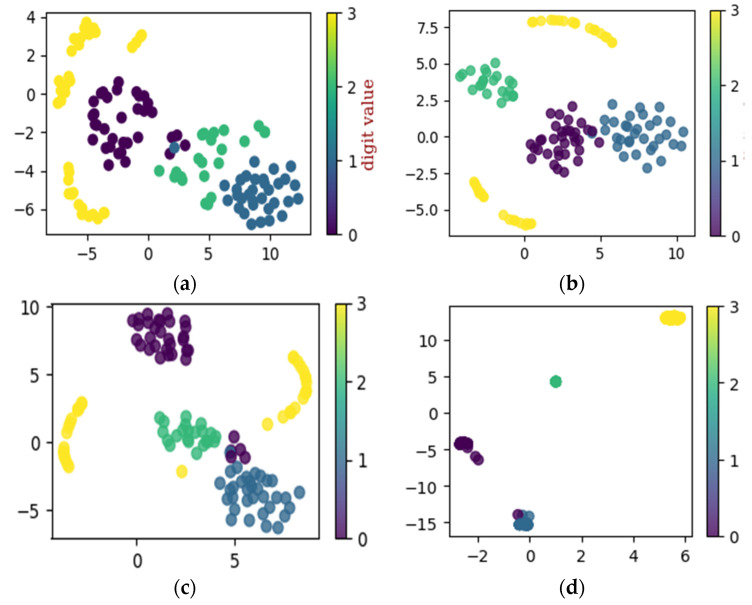
The t-SNE for self-learning ability utilizing dataset configuration D: (**a**) raw data, (**b**) convolution layer, (**c**) convolution layer, and (**d**) final classification.

**Figure 17 sensors-22-07534-f017:**
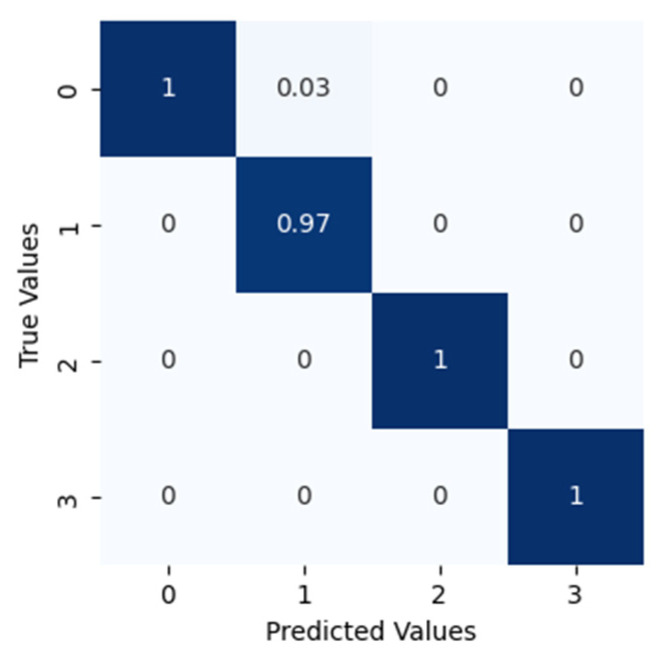
The confusion matrix for this classification.

**Figure 18 sensors-22-07534-f018:**
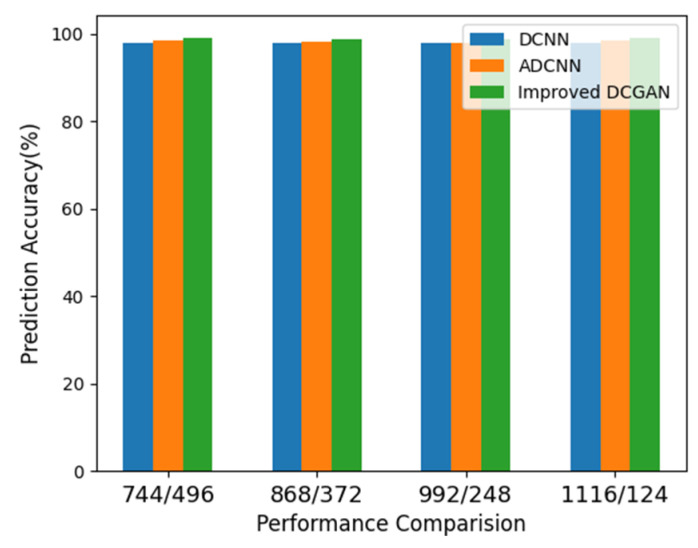
The performance of the different methods.

**Table 1 sensors-22-07534-t001:** The parameter information for the generative model.

No.	Branch 1-Layer Type	Kernel Size/Stride
1	Convolution1	3*3
2	Pooling1	2*2
3	Dropout1	0.02
4	Convolution2	3*3
5	Pooling2	2*2
6	Dropout2	0.02
7	Flatten1	32
8	Full-connectedlayer1	4

**Table 2 sensors-22-07534-t002:** The parameter information for the discriminative model.

No.	Branch 2-Layer Type	Kernel Size/Stride
1	Convolution1	3*3
2	Pooling1	2*2
3	Dropout1	0.02
4	Convolution2	3*3
5	Pooling2	2*2
7	Flatten1	32
8	Full-connectedlayer1	4

**Table 3 sensors-22-07534-t003:** Sensor information.

Devices	Specification
RCB-622B01(Vibration sensor)	Measurement range: ±490 m/s^2^
	Frequency: 0.2–15,000 Hz
	Sensor sensitivity: 100 mV/g
R151-AST (AE sensor)	Operating range: 50–400 kHz
	Resonant frequency:150 kHz (Ref in V/μbar)
	Peak sensitivity: −22 dB (Ref in V/μbar)
DAQ-type NI 9234	Dynamic range: 102 dB
	Resolution: 24-bit
	Operating temperature: −40 °C to 70 °C

**Table 4 sensors-22-07534-t004:** Prediction precision for different methods.

Data Set (Training/ Testing)	Methods	Prediction Accuracy (%)				Total Prediction Accuracy (%)
		Normal	Inner Race Fault	Outer Race Fault	Roll Race Fault	
A (744/496)	Proposed	99%	97%	100%	100%	99.0%
	ADCNN	99%	96%	98%	100%	98.5%
	DCNN	99%	94%	99%	100%	97.9%
B (868/372)	Proposed	99%	97%	99%	100%	98.9%
	ADCNN	94%	100%	98%	100%	98.3%
	DCNN	100%	91%	99%	100%	97.8%
C (992/248)	Proposed	100%	95%	100%	100%	98.9%
	ADCNN	94%	100%	99%	99%	98.0%
	DCNN	97%	94%	100%	100%	97.9%
D (1116/124)	Proposed	97%	100%	100%	100%	99.2%
	ADCNN	97%	94%	100%	100%	98.4%
	DCNN	98%	92%	100%	100%	97.8%

## Data Availability

Data will be provided upon request.

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
