# Peer review of "A Novel Image-Based Diagnosis Method Using Improved DCGAN for Rotating Machinery"

_sensors, 2022, doi:10.3390/s22197534_

Round 1

Reviewer 1 Report

In general, this is an interesting manuscript. The DCGAN architecture could be considered as the main contribution of this article to the existing state of the art. 

The authors transform bearing vibration signals into 2-D digital images, and then employ deep learning techniques for the fault type identification. 

Here are some questions to be addressed before a positive recommendation could be given for this manuscript. 

The transformation of a 1-D vibration signal into a 2-D digital image has been employed in different fault identification scenarios. The authors use the classical approach based on CWT method. Recently, such a 1-D to 2-D transformation in deep learning based bearing fault diagnosis was implemented by means of the permutation entropy [A]. The authors could discuss the advantages and disadvantages of the CWT method compared to the permutation entropy based method. 

It is very good that the authors use a standard dataset of bearing faults represented by their vibration data (the Uslan University dataset). However, this dataset is recorded on the laboratory based bearing test platform. Vibration signals recorded on machines in real world environment would be inevitably contaminated by external noise. Therefore, it is important to discuss the robustness of the proposed fault identification technique to the additive noise. 

The accuracy of the reported fault classification technique is very good. However, it is also important to discuss the sensitivity of the technique in respect to the severity of the fault. For example, the Case Western University Bearing Fault Dataset comprises vibration signals for different severity of the fault. For example, the artificial fault on the inner surface of the bearing is artificially induced by making cracks of different depths (starting from the smallest depth). The machine is assembled, the signals are measured, disassembled, measured again, and continued until the largest crack depth. Such an approach helps to explore the ability of the technique to detect defects at the early stage of their development. The authors should carefully discuss these issues in the revised manuscript too.

[A] Permutation entropy based 2D feature extraction for bearing fault diagnosis. Nonlinear Dynamics. 2020, vol.102, 1717-1731.

Reviewer 2 Report

This paper presents a new image-based diagnostic method using an improved deep convolutional generative adversarial network (DCGAN) method for fault detection and classification in rolling bearings. 

I consider that the paper is original and the research is well conducted. Anyway, I have some suggestions that may improve the clarity of the article.

 1. A flowchart of the entire research plan included in this paper should be presented, including the predictive DCGAN model.

2. Indicate the fault size for every rolling bearing element and the way they were created. In my opinion, for such large faults, the Fourier Transform is suited to detect them. Present the amplitude-frequency Fourier spectra for each faulty element and for the healthy rolling bearing state. Please comment on your choice to complicate the diagnosis method. 

3. Pay attention to wrong expressions such as " inner damaged bearings, outer damaged bearings", and replace them all over the paper with "inner race damaged bearings, outer race damaged bearings". The same objection to "inner fault" and "outer fault" expressions.

4. Check the captions of Figures 9, 11, 13, and 15. For b) one layer, and for c) one layer too?! 

5. Indicate the prediction accuracy for each fault: inner ring, outer ring, and rollers, and discuss the results.

6. A clarification is also needed for the last sentence from the Discussions section: "However, some methods about signals processing need more change in the future." What signal processing methods do the authors intend to improve and why?  
